# PKCeta Promotes Stress-Induced Autophagy and Senescence in Breast Cancer Cells, Presenting a Target for Therapy

**DOI:** 10.3390/pharmaceutics14081704

**Published:** 2022-08-16

**Authors:** Noa Rotem-Dai, Amitha Muraleedharan, Etta Livneh

**Affiliations:** The Shraga Segal Department of Microbiology, Immunology and Genetics, Faculty of Health Sciences, Ben-Gurion University of the Negev, Beer Sheva 8410501, Israel

**Keywords:** protein kinase C, PKCη, autophagy, senescence, chemoresistance, oxidative stress, ER stress, 3MA, chloroquine

## Abstract

The emergence of chemoresistance in neoplastic cells is one of the major obstacles in cancer therapy. Autophagy was recently reported as one of the mechanisms that promote chemoresistance in cancer cells by protecting against apoptosis and driving senescence. Thus, understanding the role of autophagy and its underlying signaling pathways is crucial for the development of new therapeutic strategies to overcome chemoresistance. We have previously reported that PKCη is a stress-induced kinase that confers resistance in breast cancer cells against chemotherapy by inducing senescence. Here, we show that PKCη promotes autophagy induced by ER and oxidative stress and facilitates the transition from autophagy to senescence. We demonstrate that PKCη knockdown reduces both the autophagic flux and markers of senescence. Additionally, using autophagy inhibitors such as chloroquine and 3-methyladenine, we show that PKCη and autophagy are required for establishing senescence in MCF-7 in response to oxidative stress. Different drugs used in the clinic are known to induce autophagy and senescence in breast cancer cells. Our study proposes PKCη as a target for therapeutic intervention, acting in synergy with autophagy-inducing drugs to overcome resistance and enhance cell death in breast cancer.

## 1. Introduction

Autophagy is regarded as a cellular ‘recycling factory’, removing non-functional proteins and organelles and generating the building blocks necessary for cell survival [1]. Three types of autophagy have been described so far based on the molecular machinery, morphological characteristics, and mechanisms by which intracellular components are delivered for degradation: microautophagy, chaperone-mediated autophagy (CMA), and macroautophagy (commonly termed as autophagy) [2,3]. Autophagy initiates with isolated membranes (phagophores) derived from a lipid bilayer, although their exact membranal origin in mammals is controversial. The phagophore expands to engulf intracellular cargo, such as protein aggregates, organelles, and ribosomes, thereby sequestering the cargo in a double-membrane autophagosome. The loaded autophagosome matures through fusion with the lysosome (autophagolysosome), promoting the degradation of the autophagosomal contents by lysosomal acid proteases. The byproducts of degradation are recycled back to the cytoplasm, where they can be reused [1,4,5]. During normal cellular homeostasis, autophagy functions as a primary route of degradation for damaged organelles and protein aggregates. As an intracellular self-destructive system, autophagy must be tightly regulated to adapt to different intracellular and extracellular stressors [6]. Interestingly, in cancer, autophagy plays opposing roles—under certain conditions, it has a cytoprotective effect that causes chemotherapy resistance; in others, it has a cytotoxic effect through which some compounds induce autophagy-mediated cell death [2,3,7].

Several studies identified autophagy as an effector mechanism of senescence that is important for the rapid protein remodeling required for an efficient transition from a proliferative to a senescent state (characterized by stable cell cycle arrest with an active metabolism). For example, it was demonstrated that oncogene-induced senescence (OIS) could be dependent on the prior induction of autophagy [8]. In agreement, autophagy and senescence appear to be regulated by an overlapping signaling pathway involving the generation of ROS, the activation of ATM, the induction of p53 and p21, and the dephosphorylation of pRb [9]. Interestingly, the suppression of autophagy induced apoptosis and attenuated senescence, suggesting that autophagy plays an important role in inducing or sustaining senescence [10].

A role for PKC family members in autophagy has recently emerged. PKCδ was demonstrated to activate autophagy by promoting Jun N-terminal kinase 1 (JNK1)-mediated Bcl-2 phosphorylation and the dissociation of the Bcl-2/Beclin-1 complex [11,12]. The pharmacological agents safingol [13] and oridonin [14] were shown to trigger autophagy via PKC [13,14]. The activation of PKCθ was required for ER stress-induced autophagy [15]. PKCη expression was reported to promote cellular senescence in MCF-7 cells in response to DNA damage [16]. Here, we show a role for PKCη in the induction of autophagy and demonstrate that both PKCη and autophagy are required for establishing senescence in MCF-7 in response to oxidative stress.

## 2. Materials and Methods

### 2.1. Cells

MCF-7 and MCF21.5 (MCF-7 cells inducibly expressing PKCη, previously described in [17]) were grown in Dulbecco’s Modified Eagle’s Medium (DMEM) containing 100 U/mL penicillin, 0.1 mg/mL streptomycin, 2 mM L-glutamine, and 10% Fetal Bovine Serum (Biological Industries, Beit Haemek, Israel) in a 5% CO_2_ humidified atmosphere at 37 °C. The medium for MCF21.5 cells additionally included hygromycin B (100 μg/mL), G418 sulfate (200 μg/mL) (Calbiochem, Merck, MA, USA), and tetracycline (2 μg/mL) (Sigma-Aldrich, Rehovot, Israel). The expression of PKCη was induced by the removal of tetracycline (-Tet) from the growth medium.

### 2.2. Antibodies and Reagents

Anti-PKCη (sc-215), p21 (sc-397), and p27 (sc-528) were purchased from Santa Cruz Biotechnology (Santa Cruz, CA, USA). Anti-PARP-1 (9542) was purchased from Cell Signaling Technology (CST, Danvers, MA, USA). Anti-LC3 (L8918) was purchased from Sigma-Aldrich (Sigma-Aldrich, Israel). Anti-p62 (ab56416) was purchased from Abcam (Abcam, Cambridge, MA, USA). Anti-H3K9meth was purchased from Abcam. Anti-actin was purchased from ICN (691001, ICN Biomedicals, Santa Ana, CA, USA). Anti-pPKCηSer675 was specially made from PhosphoSolutions (PhosphoSolutions, Aurora, CO, USA). The iPKCη peptide, a PKCη pseudosubstrate inhibitor (myristoylated), was purchased from Calbiochem (Cat. No. 539604). The horseradish peroxidase conjugated to donkey anti-rabbit (NA934V) or anti-mouse (NA931V) immunoglobulin was from Amersham Biosciences (Piscataway, NJ, USA).

For the inhibition of autophagy, 3-Methyladenine (3MA), a PI3K inhibitor, was purchased from Merck (Merck Millipore, Burlington, MA, USA). Chloroquine (CQ) was purchased from Sigma-Aldrich.

### 2.3. Cell Lysis and Western Blot Analysis

Cell lysis and western blot analysis were performed as described. Briefly, whole-cell extracts were prepared by lysing cells in RIPA lysis buffer containing 10 mM Tris (pH 8.0), 100 mM NaCl, 5 mM EGTA (pH 8.0), 45 mM 2-mercaptoethanol, 1% NP-40, 10 mM EGTA (pH 8.0), 50 mM NaF, and 0.1% SDS supplemented with protease inhibitors (1 mM PMSF, 10 µg/mL aprotinin, and 10 µg/mL leupeptin) and phosphatase inhibitors (1 mM sodium orthovanadate, 50 mM β-glycerol phosphate, and 5 mM sodium pyrophosphate). The lysates were incubated on ice for 30 min, sheared several times through a 21-gauge needle, and centrifuged at 13,000× *g* for 25 min at 4 °C. The protein concentrations were determined by using the Bio-Rad (Hercules, CA, USA) protein assay, and aliquots of 30–60 μg protein were prepared. The samples were resolved by electrophoresis on 10–15% polyacrylamide gels using Bio-Rad Mini-PROTEAN II cells. Proteins from the gel were electroblotted onto PVDF (Bio-Rad) in Bio-Rad Mini Trans-Blot transfer cells followed by 1 h of blocking with 3% BSA in PBS at 37 °C. The PVDF membranes were incubated sequentially with indicated primary antibodies overnight at 4 °C, followed by the HRP-conjugated secondary antibodies. Immunoreactive protein bands were detected using the ECL reagent (Biological Industries) by the GelDoc (Bio-Rad) system.

### 2.4. SA-β-Galactosidase Staining

Senescence-associated β-galactosidase (SA-β-gal) activity was determined using a previously described protocol [16]. Briefly, the cells were washed once with PBS and fixed with 0.5% glutaraldehyde for 15 min followed by 2× PBS wash supplemented with 1 mM MgCl_2_. The cells were stained in X-gal solution (1 mg/mL X-gal, 0.12 mM K_3_Fe[CN]_6_, 0.12 mM K_4_Fe[CN]_6_ in PBS at pH 6.0) overnight at 37 °C. The cells were photographed using an IX70Olympus optical light microscope. To estimate the total cell numbers, the cell cultures were stained with 10 μg/mL Hoechst 333432 (#H6024, Sigma-Aldrich, Israel) for 30 min at 37 °C before β-gal staining. Hoechst fluorescence was detected using a light source providing light at 340–380 nm; the emission was at 465 nm. The percentage of SA-β-gal-positive cells out of Hoechst-stained cells was calculated using Image J software.

### 2.5. Statistical Analysis

All statistical analyses were performed with GraphPad Prism version 9 for Windows (GraphPad Software, San Diego, CA, USA). All variables are expressed as means ± SEM. The *p*-values were calculated using an unpaired Student’s *t*-test or with a one-way ANOVA, as indicated in the figure captions.

## 3. Results

### 3.1. PKCη Enhances Autophagy Induced by ER and Oxidative Stress

The novel PKC isoform, PKCη, is an anti-apoptotic stress-induced kinase that was shown to be involved in a variety of cellular responses, such as differentiation, proliferation, and secretion [18,19]. We have previously reported that PKCη promotes cellular senescence in response to oxidative stress [16]. Our aim here was to determine whether PKCη has a role in oxidative stress-induced autophagy, since autophagy and senescence are indicated to be interconnected.

PKCη is localized in the perinuclear region, the endoplasmic reticulum (ER), and the Golgi apparatus [20], suggesting that it may have a role in ER stress-induced autophagy. ER stress is caused by the accumulation of unfolded and/or misfolded proteins in the ER lumen, resulting in an adaptive signaling pathway termed the unfolded protein response (UPR) and in the degradation of misfolded proteins through ERAD and/or autophagy. As ER stressors, we used Tunicamycin (TM) (inhibits N-acetylglucosamine (GlcNAc) phosphotransferase) and Thapsigargin (TG) (inhibits the sarco/endoplasmic reticulum ATPase (SERCA), resulting in an increased cytosolic Ca^2+^ concentration).

MCF-7 cells, inducibly expressing PKCη under the control of the tetracycline-responsive promoter (MCF21.5 cells, previously described in [17]), were treated with H_2_O_2_ in the presence or absence of chloroquine (CQ). CQ inhibits autophagy through the inhibition of autophagosome-lysosome fusion [21]. CQ causes lysosomal dysfunctions, limiting the lysosomal protein degradative capacity and thus causing the blockage of the autophagy flux, resulting in the accumulation of LC3-II in autophagosomes [22]. As shown in Figure 1A, under conditions of both oxidative stress (H_2_O_2_) and the inhibition of autophagy (CQ), the induced expression of PKCη in MCF-7 cells (upon the removal of tetracycline (-Tet)) depicted higher levels of the autophagic marker LC3-II compared to the control cells (+Tet). When unstressed cells were treated only with CQ, the level of LC3-II was not altered by PKCη expression. Our results show that PKCη enhances autophagy only under oxidative stress conditions, in line with its role as a stress-induced kinase [23].

The effects of PKCη in enhancing autophagy in response to ER stress by TM or TG were similar to those observed by oxidative stress. As shown in Figure 1B,C, when cells were treated with TM (10 μg/mL) and TG (100 nM) for 24 h, LC3-II levels were similar in PKCη-expressing cells (-Tet) compared to control cells (+Tet). However, when the autophagic flux was inhibited by CQ in these TM- and TG-treated cells, the levels of LC3-II were higher in PKCη-expressing cells (-Tet) compared to control cells (+Tet), suggesting that PKCη enhances autophagy under ER stress conditions.

Another marker used to monitor the autophagic flux is p62, also called sequestosome 1 (p62/SQSTM1). p62 possesses a short LC3 interaction region (LIR) that facilitates direct interaction with LC3 and GABARAP family members and causes p62 to be specifically degraded by autophagy. Because its degradation is dependent on autophagy, the level of p62 increases when autophagy is inhibited [24]. As depicted in Figure 1A,B, in response to oxidative stress and ER stress by TM, the accumulation of p62 levels was observed in PKCη-expressing cells (-Tet) compared to control cells (+Tet) upon the inhibition of the autophagic flux, supporting the idea that autophagy was promoted by the overexpression of PKCη. 

The activation and stability of PKC family members are dependent on post-translational regulation such as phosphorylation and translocation [25]. PKCη contains three conserved phosphorylation sites: the activation loop (Thr513), turn motif (Thr655), and hydrophobic motif (Ser675) [26]. We have previously shown that phosphorylation on the hydrophobic motif Ser675 of PKCη is increased in response to stress by the chemotherapeutic drug etoposide [27]. As depicted in Figure 1D,E, the levels of phosphorylation on the Ser675 of PKCη were increased following TM and H_2_O_2_ treatment, demonstrating that PKCη is activated.

### 3.2. PKCη Knockdown Reduces Autophagy 

MCF-7 cells stably transfected with shRNA-PKCη constructs (shPKCη3-5 and shPKCη2-2) and control plasmid (shScr5-3) (previously described in [16]) were treated with/without H_2_O_2_ in the presence/absence of CQ (Figure 2A). Treatment with both H_2_O_2_ and CQ resulted in lower levels of LC3-II in PKCη-knockdown cells (shPKCη3-5 and shPKCη2-2) compared to control cells (shScr5-3), suggesting that PKCη knockdown reduced the autophagic flux in response to oxidative stress.

To confirm that the knockdown of PKCη results in the inhibition of autophagy, we examined the effects of a PKC-inhibitory peptide (iPKCη) developed to specifically inhibit PKCη kinase activity. The peptide iPKCη itself did not affect LC3-II levels: however, its presence diminished H_2_O_2_-induced LC3-II levels (Figure 2B). Taken together, our results demonstrate that PKCη enhances autophagy in response to oxidative stress.

### 3.3. Inhibition of Autophagy Attenuates the Induction of Senescence by PKCη

Our current study shows a role for PKCη in promoting autophagy under oxidative stress. To investigate the transition from autophagy to senescence under the conditions of oxidative stress, we inhibited autophagy using 3MA, followed by assessing SA-β-gal and other senescence markers. 3MA is an inhibitor of the activity of class III phosphatidylinositol 3-kinase (PI3K type III) that prevents the formation of autophagosomes, thereby blocking the autophagic process [28]. As shown in Figure 3A,B, the addition of 3MA to the H_2_O_2_-treated MCF-7 cells decreased SA-β-gal staining. The expression levels of the senescence markers, the cell cycle inhibitor p21, and the histone H3K9me2/3 were reduced in H_2_O_2_-treated cells upon 3MA application (Figure 3C). Furthermore, the inhibition of autophagy by 3MA promoted apoptotic cell death in H_2_O_2_-treated cells, as indicated by the increased levels of cleaved PARP-1 (Figure 3D). Treatment with CQ increased LC3-II levels and enhanced PARP-1 cleavage compared to control non-treated cells (Figure 3D). Our data suggest that autophagy is required to protect MCF-7 cells from stress-induced cell death.

We have previously demonstrated a role for PKCη in promoting senescence [16]. To investigate the role of PKCη in regulating the transition from autophagy to senescence, we employed the PKCη-knockdown cells described above. As shown in Figure 3E,F, the percentage of SA-β-gal-positive cells following oxidative stress was lower in PKCη-knockdown cells (shPKCη2-2) compared to control cells (shScr5-3), in accordance with the increased expression of the senescence marker p21 in these cells (Figure 3G), further demonstrating that PKCη promotes senescence. The inhibition of autophagy using CQ significantly reduced SA-β-gal-positive cells in PKCη-expressing MCF-7 cells (shScr5-3) (Figure 3E,F), which was also accompanied by a reduced expression of the senescence marker p21. The oxidative stress-induced autophagic flux was also reduced in PKCη-knockdown cells, as indicated by lower levels of LC3-II compared to scrambled control cells (Figure 3G). Taken together, our results suggest that PKCη promotes oxidative stress-induced senescence via the upregulation of autophagy.

## 4. Discussion

One of the major obstacles in cancer therapy is the emergence of chemoresistance. Recently, autophagy was reported as one of the mechanisms that promote resistance in cancer cells (e.g., in HER2-positive breast cancer) [29]. Autophagy functions to remove oxidized proteins and damaged mitochondria accumulating in cells [30], and it is frequently linked to the induction of cellular senescence [9]. In this study, we show that PKCη promotes autophagy in response to ER and oxidative stress. Moreover, we also show a role for PKCη in the transition from autophagy to senescence in stress-induced cells by demonstrating that its knockdown reduces both the autophagic flux and the induction of senescence.

Using both PKCη-overexpressing MCF-7 cells and cells with endogenously reduced PKCη expression by shRNA knockdown, we demonstrate that PKCη enhances the autophagic flux upon stress. Additionally, the inhibition of PKCη kinase activity in MCF-7 cells using a specific PKCη inhibitor (iPKCη) resulted in lower levels of LC3-II, suggesting that PKCη activity is required for enhancing autophagy.

Autophagy is currently considered an important mechanism for intervention in anticancer therapy to abrogate cancer resistance [2,3,7,29]. Thus, to design new antitumoral therapies that target stress-induced autophagy, it is important to understand the regulatory pathways modulating autophagy. Our results, demonstrating that PKCη promotes autophagy induced by ER stress and oxidative stress in MCF-7 cells, may be of therapeutic relevance in breast tumors expressing high levels of PKCη. Targeting this signaling pathway could reduce resistance to anti-cancer therapy, since we have already established that PKCη promotes chemotherapeutic resistance in breast cancer cells treated with DNA-damaging drugs [16].

Several studies have demonstrated that autophagy is followed by senescence in response to oxidative stress [31,32]. PKCη was previously shown to promote cellular senescence induced by DNA damaging agents (H_2_O_2_ or Etoposide), as demonstrated by the increased expression of senescence markers (p21 and pRb) and the elevated senescence-associated secretory phenotype (SASP) [16]. Another study presented evidence that the induction of autophagy by p38α protects U2OS cancer cells from doxorubicin-induced apoptosis by promoting senescence [33]. Here, we show that PKCη expression promotes the autophagy and senescence induced by oxidative stress and that the inhibition of either autophagy or PKCη expression reduces cellular senescence. Several studies have demonstrated that the interference/inhibition of autophagy compromises the generation of senescent cells; however, whether senescence is entirely dependent on prior autophagy is still under debate. Our study supports the role of autophagy as a pro-senescence mechanism required to induce senescence in MCF-7 breast cancer cells.

In the present study, we show that PKCη is involved in the cellular response to oxidative stress through its role in autophagy regulation and the subsequent induction of senescence. Upregulated PKCη expression in stressed cells [34], exhibiting activated survival pathways such as autophagy and senescence, implies that targeting PKCη could act in synergy with drugs used in breast cancer therapy to induce oxidative stress and autophagy, such as Lapatinib, Gemcitabine, Tamoxifen, Trastuzumab, etc. [3,29,35]. Thus, targeted therapy against PKCη could restrict breast cancer cell survival and chemoresistance in response to drug-induced autophagy.

The idea that PKCη could serve as a target for therapy in breast cancer is further supported by our recent report of an upstream open reading frame (uORF)-encoded micropeptide in the 5′UTR on the mRNA coding for PKCη that acts in cis as a kinase inhibitor for PKCη and in trans for other novel PKC family members (PKCδ, ε, θ), but not for classical or atypical PKCs [36]. The micropeptide contains a pseudosubstrate-like motif, which is characteristic of all PKCs that inhibit their kinase activity. The treatment of breast cancer cells by this micropeptide inhibited their proliferation and migration and synergized with chemotherapy by interfering with the response to DNA damage. Furthermore, it suppressed tumor progression, invasion, and the metastasis of breast cancer xenograft mice models. Its role in suppressing autophagy and senescence in combination with chemotherapy is currently being investigated. 

## Figures and Tables

**Figure 1 pharmaceutics-14-01704-f001:**
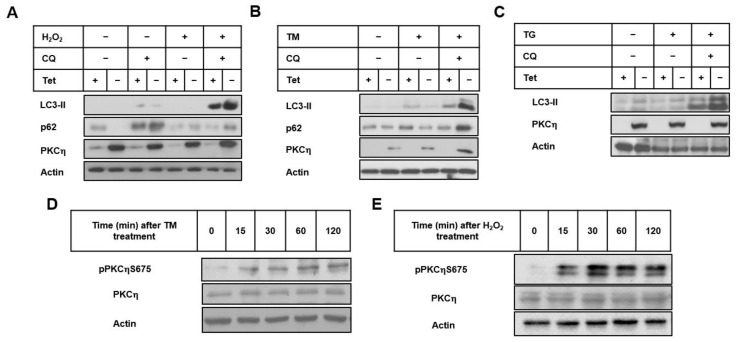
PKCη enhances autophagy induced by ER and oxidative stress. (**A**) Sub-confluent MCF21.5 cells (MCF-7 cells inducibly expressing PKCη) were grown in the presence/absence of tetracycline (2 μg/mL) for 48 h. Cells were incubated with CQ (10 µM) for 1 h in serum-free medium, followed by H_2_O_2_ (150 µM) treatment for 2 h. Fresh growth medium was replaced for 24 h. (**B**,**C**) Sub-confluent MCF21.5 cells were grown as described in (**A**). After CQ treatment, Tunicamycin (TM, 10 μg/mL) or Thapsigargin (TG, 100 nM) were added for 24 h, respectively. Whole-cell lysates were prepared and subjected to immunoblotting with the indicated antibodies. (**D**) Sub-confluent MCF-7 cells were treated with TM (10 μg/mL or (**E**) H_2_O_2_ (150 μM) for the indicated time points. The phosphorylation of Ser675 on PKCη was detected using a specific antibody. Actin was used as the loading control. Results are representative of three independent experiments.

**Figure 2 pharmaceutics-14-01704-f002:**
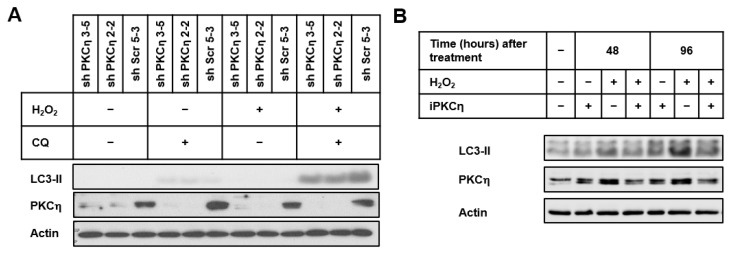
PKCη knockdown reduces oxidative stress-induced autophagy. (**A**) Sub-confluent MCF-7 stably transfected with shRNA constructs (shPKCη3-5, shPKCη2-2) or scrambled control plasmid (shScr5-3) were treated with CQ (10 μM) for 1 h in serum-free medium, followed by H_2_O_2_ (150 μM) for 2 h. Fresh growth media were replaced for 96 h. (**B**) Sub-confluent MCF-7 cells were treated with an inhibitory peptide for PKCη (iPKCη) for 4 h in serum-free medium, followed by treatment with H_2_O_2_ (150 μM) for 2 h. Fresh growth medium was replaced for 48 h and 96 h. At the end of the experiments, whole-cell lysates were prepared and subjected to immunoblotting with indicated antibodies. Actin was used as the loading control. Results are representative of three independent experiments.

**Figure 3 pharmaceutics-14-01704-f003:**
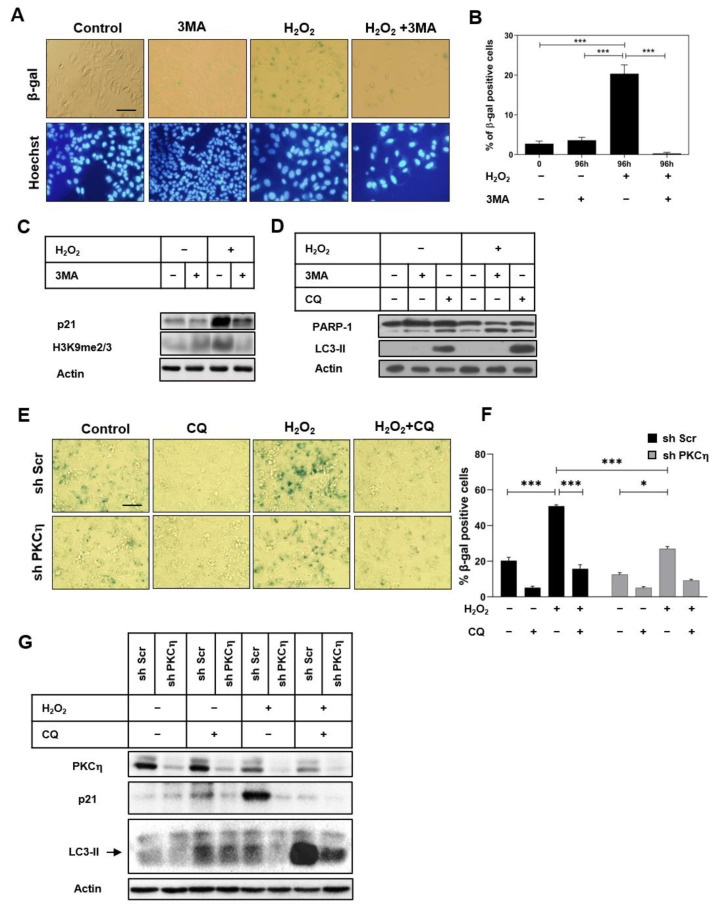
Inhibition of autophagy attenuates the induction of senescence by PKCη. (**A**) Sub-confluent MCF-7 cells were treated with 3MA (1 mM) for 1 h in serum-free medium, followed by the addition of H_2_O_2_ (150 μM) for 2 h. Fresh growth medium was added for 96 h. Cells were stained for SA-β-gal, and the nuclei were stained with Hoechst. Images were taken by a fluorescence inverted microscope. Scale bar equals 50 μm. (**B**) Quantification of SA-β-gal-positive cells was performed as described in the Materials and Methods section. (**C**,**D**) Sub-confluent MCF-7 were grown and treated as described in (**A**). Whole-cell lysates were prepared and subjected to immunoblotting with indicated antibodies. (**E**) Sub-confluent MCF-7 stably transfected with shRNA constructs (shPKCη2-2) or control plasmid (shScr5-3) were grown for 48 h and were treated with CQ (10 μM) for 1 h, followed by H_2_O_2_ (150 μM) in serum-free medium for 2 h. Fresh growth medium was added for 96 h. Cells were stained for SA-β-gal, and images were taken by a fluorescence inverted microscope. (**F**) Quantification of SA-β-gal positive cells was determined as described in the Materials and Methods section. (**G**) Whole-cell lysates were subjected to immunoblotting with indicated antibodies. Actin was used as the loading control. Results are representative of three independent experiments. Statistical analysis was performed using one-way ANOVA; * *p* < 0.05, *** *p* < 0.001.

## Data Availability

The data that support the findings of this study are available from the corresponding author on reasonable request.

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
