# Peer review of "PKCeta Promotes Stress-Induced Autophagy and Senescence in Breast Cancer Cells, Presenting a Target for Therapy"

_pharmaceutics, 2022, doi:10.3390/pharmaceutics14081704_

Round 1

Reviewer 1 Report

The manuscript entitled, PKCeta promotes stress-induced autophagy and senescence in breast cancer cells, presenting a target for therapy, by Rotem-Dai and Muraleedharan et al reports on the role of novel Protein Kinase C isoform, PKCeta (PKCη) in triggering autophagy and senescence in breast cancer under stress conditions. The manuscript, which is submitted as a 'Brief Report", is an interesting work that is well-written and is suitable for publication in Pharmaceutics, under the special issue-"Cancer therapy resistance: choosing kinase inhibitors". The study demonstrates a novel role for PKCη in modulating autophagy under stress leading to senescence. However, the authors shall address a few comments/ specific points which are listed below for their consideration.

            1 In results 3.3, in Figure 3 panel G, the authors must add an immunoblot of senescence marker (like p21), which will strengthen their conclusion.

    2.       The study emphasizes utilizing PKCη inhibitory peptides which could potentially be used in combination with autophagy inducers. This looks like a promising strategy. Very recently from this lab, a study demonstrated the ability of a uORF-encoded peptide as a novel class PKC inhibitor, which possesses anti-tumor abilities (against breast cancer) and synergistic effects with chemotherapy via the down-regulation of the kinase PKCη. The authors shall cite this paper in this current study and briefly discuss the possibilities of the utilization of PKCη-specific, small-kinase inhibitory molecules, in the context of autophagy and senescence.

 3.      Though commercially-obtained, the modifications and amino acid composition of iPKCη peptide should be listed in the Reagents section.

4.     In results 3.3, correct SA-b-gal to SA-b-gal in lines 181, 189, 191.

5.   Line 33 (chaperone-mediated), line 77 (peroxidase-conjugated), line 186 (stress-induced), and line 214 (DNA-damaging), add hyphens to these words.

Author Response

Response to Reviewer 1 Comments

Point 1: In results 3.3, in Figure 3 panel G, the authors must add an immunoblot of senescence marker (like p21), which will strengthen their conclusion.

Response 1: As requested by the reviewer, the immunoblot of p21 was added to Figure 3 panel G. The data obtained are described in the Results section (lines 194-201).

Point 2: The study emphasizes utilizing PKCη inhibitory peptides which could potentially be used in combination with autophagy inducers. This looks like a promising strategy. Very recently from this lab, a study demonstrated the ability of a uORF-encoded peptide as a novel class PKC inhibitor, which possesses anti-tumor abilities (against breast cancer) and synergistic effects with chemotherapy via the down-regulation of the kinase PKCη. The authors shall cite this paper in this current study and briefly discuss the possibilities of the utilization of PKCη-specific, small-kinase inhibitory molecules, in the context of autophagy and senescence.

Response 2: Our recent study on a uORF-encoded peptide was cited and discussed in lines 252-259.

Point 3: Though commercially obtained, the modifications and amino acid composition of iPKCη peptide should be listed in the Reagents section.

Response 3: The complete information on iPKCh was added to the list of Antibodies and Reagents: “The iPKCh peptide, a PKCh pseudosubstrate inhibitor (myristoylated) was purchased from Calbiochem (Cat. No. 539604)”. No sequence information is provided by the company. According to the literature, the sequence is Myr-TRKRQRAMRRRVHQING-NH2.

Point 4. In results 3.3, correct SA-b-gal to SA-b-gal in lines 181, 189, 191.

Response 4: Was corrected (highlighted in yellow)

Point 5. Line 33 (chaperone-mediated), line 77 (peroxidase-conjugated), line 186 (stress-induced)and line 214 (DNA-damaging), add hyphens to these words.

Response 5: Was corrected (highlighted in yellow)

Reviewer 2 Report

The authors have previously reported that PKCη can confer resistance in breast cancer cells against chemotherapy by inducing senescence. Here they show that PKCη promotes autophagy induced by ER and oxidative stress and facilitates the transition from autophagy to senescence. Three experiments were carried out to prove this viewpoint. They proposes PKCη as a target for therapeutic intervention, acting in synergy with autophagy-inducing drugs, to overcome resistance and enhance cell death in breast cancer.

There are several questions that the author should to explain:

1CQ was used as autophagy inhibitors, the level of LC3-II in the cells treated with CQ should decrease or increase? CQ was used as autophagy inhibitors, the level of LC3-II was not altered, why?

2、  In the line 127, -Tet and +Tet should be explained carefully, in the line 129 “When cells…”, the cells were -Tet or +Tet? From line 124-130, your explanation was confused that could not support you conclusion, please rephrase.

3The paper has format problems, please check it carefully.

Author Response

Response to Reviewer 2 Comments

Point 1: CQ was used as autophagy inhibitors, the level of LC3-II in the cells treated with CQ should decrease or increase? CQ was used as autophagy inhibitors, the level of LC3-II was not altered, why?

Response 1: Chloroquine (CQ) inhibits autophagy through the inhibition of autophagosome-lysosome fusion. CQ is a weakly basic drug that accumulates in lysosomes, thereby increasing lysosomal pH and limiting their protein degradative capacity. CQ inhibits the autophagic flux and enables the accumulation of LC3-II in autophagosome membranes. In our experiments, the induction of autophagy by ER or oxidative stress increased LC3-II, which was evident by the accumulation of LC3-II in the presence of CQ. PKCh expression further increased LC3-II in these stressed cells (Fig. 1, 2), demonstrating that they have a higher flow of autophagy. In unstressed cells that were treated by CQ alone, a slight increase in LC3-II was often exhibited, which was not affected by PKCh expression. This is in line with PKCh being a stress-induced kinase.

We agree with the reviewer that this part was not written clearly and was therefore rephrased (lines 127-131).

Point 2:  In the line 127, -Tet and +Tet should be explained carefully, in the line 129 “When cells…”, the cells were -Tet or +Tet? From line 124-130, your explanation was confused that could not support you conclusion, please rephrase.

Response 2: This paragraph was rephrased (lines 123-131).

Point 3: The paper has format problems, please check it carefully.

Response 3: The format was carefully checked and amended.
